# Family Caregiver Strain and Challenges When Caring for Orthopedic Patients: A Systematic Review

**DOI:** 10.3390/jcm9051497

**Published:** 2020-05-16

**Authors:** Umile Giuseppe Longo, Maria Matarese, Valeria Arcangeli, Viviana Alciati, Vincenzo Candela, Gabriella Facchinetti, Anna Marchetti, Maria Grazia De Marinis, Vincenzo Denaro

**Affiliations:** 1Department of Orthopedic and Trauma Surgery, Campus Bio-Medico University, Via Alvaro del Portillo, 200, 00128 Trigoria, Rome, Italy; v.candela@unicampus.it (V.C.); denaro@unicampus.it (V.D.); 2Research Unit Nursing Science, Campus Bio-Medico di Roma University, 00128 Rome, Italy; m.matarese@unicampus.it (M.M.); arcangeli.valeria@libero.it (V.A.); vivianaalciati@gmail.com (V.A.); g.facchinetti@unicampus.it (G.F.); a.marchetti@unicampus.it (A.M.); m.demarinis@unicampus.it (M.G.D.M.)

**Keywords:** caregiver, orthopedic disease, caregiver strain, hip, knee, shoulder, caregiver stress, dyads

## Abstract

Background: Caregivers represent the core of patients’ care in hospital structures, in the process of care and self-care after discharge. We aim to identify the factors that affect the strain of caring for orthopedic patients and how these factors are related to the quality of life of caregivers. We also want to evaluate the role of caregivers in orthopedic disease, focusing attention on the patient–caregiver dyad. Methods: A comprehensive search on PubMed, Cochrane, CINAHL and Embase databases was conducted. This review was reported following PRISMA statement guidance. Studies were selected, according to inclusion and exclusion criteria, about patient–caregiver dyads. For quality assessment, we used the MINORS and the Cochrane Risk of BIAS assessment tool. Results: 28 studies were included in the systematic review; in these studies, 3034 dyads were analyzed. Caregivers were not always able to bear the difficulties of care. An improvement in strain was observed after behavioral interventions from health-care team members; Conclusions: The role of the caregiver can lead to a deterioration of physical, cognitive and mental conditions. The use of behavioral interventions increased quality of life, reducing the strain in caregivers of orthopedic patients. For this reason, it is important to consider the impact that orthopedic disease has on the strain of the caregiver and to address this topic.

## 1. Introduction

Orthopedic surgery is one of the most commonly performed surgeries worldwide [1,2]. Patients undergoing orthopedic surgery can experience difficulties in the management of post-surgical symptoms and physical limitations [3]. Orthopedic patients may experience barriers such as difficulties with Activities of Daily Living (ADL) [4,5], and problems returning to post-surgical lives. For these reasons, the role of the caregiver is of paramount importance in supporting dependent people both in simple and complex activities [6]. Although they are often family members without formal training, they take part in the activities of daily care, offering emotional support to the patients and replacing, in whole or in part, the physically dependent patients in the ADLs. Additionally, they monitor the patient’s care pathway, managing the symptoms and taking on the family responsibilities previously managed by the patients [7]. All these factors contribute to increasing the caregivers’ workload and they could affect caregivers’ quality of life, both pre to post the patients’ orthopedic surgeries [8]. The poor physical conditions of patients are associated with a decrease in the quality of life of the caregiver and an increase in stressors, due to all the caregivers’ responsibilities; the caregiver can also present physical problems due to the effort involved in helping the patient to move. It is therefore important to enhance the caregiver’s safety to improve his/her physical condition [9]. Caregiving is emotionally and cognitively demanding, and literature indicates that caregivers’ overall health is adversely altered [10].

The term “dyad” refers to the relationship between patient and caregiver, who are involved physically and emotionally. The dyad is an important foundation for problem identification and problem-solving included in the orthopedic patient process, from the pre-operative period to the post-operative period. Humor, reassurance, and empathy are also important factors in this dyad relationship. There is a need to enhance patient–caregiver dyad research, by studying the relationship from pre-operative to follow-up to evaluate the changes in the patient’s outcomes, but also in the caregiver’s psychological and cognitive sphere. There is a need to identify the type of supportive relationship that is established with informal caregivers to offer them an appropriate educational plan. Furthermore, it is necessary to evaluate if the degree of instruction received during the hospitalization period and the knowledge of informal caregivers about the disease, is sufficient for them to be able to face a functional recovery process in the postoperative period in order to maintain the role of caregiver after patients’ surgery [11].

The present systematic review aims to identify, analyze and synthesize the studies on the role of informal caregivers’ strain and difficulties when caring for orthopedic patients, focusing attention on the patient–caregiver dyad, during the pre-operative to post-operative period.

## 2. Materials and Methods

A literature review was performed and reported following PRISMA statement guidance [12]. Preliminary searches of main databases could not find any existing or ongoing systematic reviews on caregiver strain or difficulties of caring for orthopedic patients. 

### 2.1. Eligibility Criteria and Search Strategy

Key words and combinations of key words were used to search the electronic databases and were organized according to the Population Intervention Comparison Outcome (PICO) model as follows.

*Study*: original studies with different study designs; English language; recent studies (from 2003 to 2020).

*Pa**r**ticipants***:** patient–caregiver dyads, orthopedic patients and informal caregivers.

*Interventions*: educational interventions, home-care rehabilitations, emotional and social supports. 

*Outcome measures*: The primary outcome of the review is the caregivers’ role in the orthopedic patients’ functional recovery, the caregivers’ knowledge to manage orthopedic disease symptoms, the caregivers’ strain, and the dyads’ quality of life; the secondary outcome is the impact that orthopedic disorders have on quality of life, on stress level and on the psychological and physical status of patients.

A comprehensive search of the databases PubMed, Medline, Cochrane, CINAHL, and Embase databases was conducted since the inception of the database to March 2020 with the English language constraint. To ensure a comprehensive search, facet analysis, necessary to identify the key terms to be used in the search strategy, was carried out. Keywords were combined using the Boolean operators “AND” and “OR”. The search strategy was iterative and flexible within the limits of the search engines of the individual databases.

The following medical subject heading (MeSH) keywords and free terms were used for the search: caregiver, spouse, orthopedic disease, orthopedic, caregiver burden, hip, knee, shoulder, elbow, wrist, hand, humerus, femur, patella, spine, ankle, foot, caregiver stress, patient–caregiver dyads. Search strategies were checked by two reviewers (VC and GF). The exclusion criteria included: formal caregivers, reviews, books, patients and caregivers without a relation. Further details about search strategies are in Appendix A.

### 2.2. Study Selection and Data Collection

Two researchers (VAr and VAl) independently reviewed all studies (title, abstract and full text) that met the inclusion criteria and extracted relevant data. Disagreements were resolved by a discussion among the reviewers.

We included observational studies, prospective studies, cohort studies, mixed studies, pre–post quasi-experimental designs, randomized controlled trials, descriptive cross-sectional studies, prospective longitudinal cohort studies, non-randomized trials, qualitatively focused ethnographic approaches and retrospective analyses. The studies included articles published from 2003 to 2020. Disagreement regarding the exclusion and inclusion criteria were decided by the senior reviewer (VD).

### 2.3. Quality Assessment 

Two reviewers independently evaluated (VAr/VAl) the potential risk of bias of the studies included using MINORS [13], a methodological index for non-randomized studies, and the Cochrane Risk of Bias Tool [14] for randomized controlled trials.

The MINORS items were scored as 0 if not reported, 1 when reported but inadequate, 2 when reported and adequate. The global ideal score was 16 for non-comparative studies and 24 for comparative studies.

The Cochrane Risk of Bias Tool assessed randomized controlled trials with the following criteria: selection, performance, detection, attrition, reporting and other biases. Each criterion was evaluated assigning zero for low risk, one point for unclear risk, and two points for high risk of bias. The potential total score range was 0–14, in which a low score indicated a higher quality level, and a high score indicated lower quality. Based on this score, an overall score of 0–1 shows high quality, an overall score of 2–3 shows moderate quality, and an overall score of >3 shows low quality [15].

### 2.4. Data Synthesis and Analysis

Data were extracted and synthesized through Microsoft Excel. Several data were extracted, and they concern outcome measures; authors and year; study design; orthopedic disease; aim of each study follow-up period; the relationship between patient and caregiver; number of patients and caregivers for each study; data findings; and study conclusion.

Data analysis was done using the description of the study and patient and intervention characteristics. Categorical variable data were reported as percentage frequencies. Continuous variable data were reported as mean values, with the range between the minimum and maximum values.

## 3. Results

The selection process is illustrated in Figure 1. The search strategy yielded 61 articles. After duplicate removal and title, abstract and full texts review, 28 studies were evaluated for methodological quality and were eligible for the review.

### 3.1. Study and Patient Characteristics

A total of 3034 patients–caregiver dyads were reviewed (Table 1).

According to the aim of the review, the studies included analyzed the patient–caregiver dyad in the orthopedic disease context. It was found that the majority of the studies included (75%) analyzed dyad characteristics in hip disease (hip fracture, hip arthroplasty, hip deformity). To a lesser extent, 11% of the studies considered patients affected by knee orthopedic disease (knee fracture, knee arthroplasty, patella fracture) and 14.3% with backbone conditions (spinal arthrodesis, scoliosis, spine deformity, and cord injury) (Table 2).

The topics of the studies included in the systematic review make it possible to evaluate the caregiver’s role in the orthopedic patients’ functional recovery, dealing with knowledge to manage symptoms and orthopedic disease in 8 studies, caregiver strain in 15 studies and dyads’ quality of life in 5 studies.

In the studies evaluating the quality of life of the dyad, 7.14% of the studies affirmed that the patient’s quality of life was improved due to the assistance provided by the informal caregiver. Despite the improvement in patients’ quality of life, their quality of functional recovery does not improve, and the 14.2% of studies state that it is influenced by caregivers’ psychological factors. 

Regarding the caregivers’ quality of life, 7.14% of the studies say the decrease in caregivers’ quality of life is related to the increasing intensity of care and caregiver strain [14]. 

The studies analyzing caregiver strain report that the factors affecting caregiver difficulties are increased recovery time (21.4%), complication and symptom management (14.2%), financial resources (10.7%), functional level reduction (10.7%), bad healthcare experiences (7%), any trusting relationships (7%), patients’ and caregivers’ age (7%), poor social support (7%), poor self-efficacy (7%), transport (3.5%), and rural environment (3.5%). Two studies identified a significant correlation between caregiver strain and caregiver characteristics such as age [16,17] and gender [17].

The studies included in the review used some different follow-up periods: the pre-operative period (12.5%) and 2 weeks (4.6%), 1 month (18.2%), 3 months (20.4%), 6 months (23.3%), and 1 year (21%) after surgery. The follow-up period analysis was useful for analyzing the caregivers’ strain duration. In studies that analyzed the duration of caregiver strain, fourteen percent of studies reported that caregiver strain lasted for one year, 10.7% for six months, 7.1% for one month and 3.5% for two years.

Regarding the studies that analyzed knowledge to manage symptoms, they reported that pre-operative education was fundamental to improve the management of symptoms. Pre-operative education and postoperative social support reduced caregiver strain in 46.5% of studies. 

To understand the relationship between the informal caregivers and the patients that form the dyads, studies reported that the major of informal caregivers were patients’ relatives. The most common relationships between the primary caregiver and the care recipient in this review included spouses (22.1%), daughters (8.7%), sons (6.7%), daughters-in-law (5.2%), and others, including partners, mothers, grandchildren and siblings (9.2%) (Table 3).

### 3.2. Intervention Characteristics 

The most common outcome measures observed in original studies according with the aim of this review were Caregiver Strain Index (CSI), utilized in 32% of the studies recruited [18,19,20,21,22,23,24,25]; Zarit Burden Interview (ZBI), in 7.14% [26,27]; Mini-Mental Test (M-MT) in 7.14% [22,27]; and Health-Related Quality of Life score (HRQL) in 14.2% of the studies [27,28,29,30].

Participants’ knowledge of the intervention and post-surgical management was tested via Knowledge Expectations of significant others (KEso) and Received Knowledge of significant others (RKso) [16]; confusion assessment method (CAM) and family version (FAM-CAM) [7] were used to measure empowering by knowledge.

Measures for general health status included the Short Form 36 Health Survey (SF-36) [18,22,31,32], the Functional Independence Measure Score (FIM) [27]; the Short Physical Performance Battery (SPPB) [27]; Time Up and Go Test (TUG) [18,19,27]; International Fitness Scale (FIS) [27]; The University of California, Los Angeles Activity Scale (UCLA) [29]; The Blaylock Risk Assessment Screening Score (BRASS) [37]; and the italics Self-Efficacy Scale (GSE) [21,26].

ADLs for patients and caregivers were assessed using the Barthel index [18,22,32]; the Reintegration to Normal Living Index (RNLI) [20]; and the Family Function Rating Scale [21]. The Visual Analogue Scale for Pain (VAS) [27,33] was the scale most often used to evaluate symptoms, as well as depression scales [24,27].

### 3.3. Quality Assessment

Most of studies included in this review (N = 27; 96.4%) were evaluated with MINORS. Of these, one study (3.57%) had low risk of bias and 26 (92.8%) had high risk of bias. The only RCT (Crotty, 2003) in this review had moderate quality due to insufficient details about the double-blinding procedure.

## 4. Discussion

This review aimed to synthesize the studies on the role of informal caregiver strain and difficulties when caring for orthopedic patients, focusing attention on the patient–caregiver dyad, from the pre-operative period to the post-operative period. Family caregivers are a relatively unused resource as a way to identify early symptoms and complications in orthopedic patients and to improve health outcomes for orthopedic patients. When pre-operative education is performed, family caregivers can apply their knowledge by acting on early recognition of symptoms [3,7].

Learning about diseases drives caregivers to satisfaction and to learn strategies to help patients, also with emotional support. Extensive efforts have been made to understand the strain felt by caregivers of patients with orthopedic disease. Orthopedic caregivers without enough information had less security in dealing with the patients’ disease than those who received an appropriate education from health care team members: this is a finding that was consistent with the stress and coping model of caregiving [29].

Caregiving and poor quality of life can relate bidirectionally. In the dyad, concerns are about mobility, pain, self-care, support from the caregiver and discharge on the same day regarding recovery expectations, and drugs and their effect on postoperative recovery [34]. Orthopedic patients feel more pain and greater difficulties with physical activity; this leads to an increase in the caregiver’s workload, and several studies underestimate this result [35]. The impact of the orthopedic disease on dyads is crucial for patients’ outcomes and caregiver strain. The physical outcomes of patients and their functional recovery, and also the impact of caregiver assistance on care transition, have been studied more frequently [36].

In this context, the formal rehabilitation program has a key role in restoring patient autonomy and also involving the caregiver to reduce and improve rehabilitation times [38,39]. When the patients’ formal post-surgery rehabilitation program is insufficient, the supply of a family caregiver is fundamental. For example, it could be necessary to improve the information and caregivers’ education about the management of symptoms. A focus on the orthopedic patient and caregiver education regarding the clinical post-operative problems, potential risks involved, and patient and caregiver roles to improve the caring process is necessary.

In many of the studies recruited, caregiver strain is one of the main topics. The difficulty of assistance is often not recognized, and it can lead to mental and physical problems, but can also have a strong impact on social life [35]. Caregiver stress increases in the immediate postoperative period, while decreasing a lot after a year after surgery [25]. Especially in patients undergoing hip replacement, the caregiver strain is very high, which is why it is important to ease caregiver stress and increase the quality of care throughout the functional recovery period, including the periodic follow-up [24]. In most of the studies analyzed, the follow-up period of the dyad is approximately one year. This means that it cannot be said that the interventions implemented on the dyad provide long-term benefits and that the outcomes are valid.

Concerning the strain of care [30], we have found that caregivers showed disorders in the cognitive and physical spheres of their daily lives during the treatment period, from the pre-operative to post-operative period, leading to a reduction in their quality of life. 

This review showed the importance of information and education before and after orthopedic surgery, to limit the functional restriction of patients by scared caregivers and to analyze its influence on functional recovery. The patient’s functional level, quality of life, physical performance, pain, caregiver strain, and their the emotional and cognitive state should be evaluated, and perception of physical state should be assessed [40]. A study conducted in 2010 aimed to investigate the causes of stress attributed to the caregivers of patients with orthopedic disease [23]. The caregiver tension can result from changes in the patient’s physical and cognitive states, and the sustained role during the patient’s daily life activities. Several factors can contribute to caregiver stress, including financial strain, which is one of the most significant causes of caregiver stress. The financial problems arise from the need to incur medical expenses, rehabilitation, and transportation. This can be added to the physical and mental stress of the caregiver.

Post-operative recovery of patients was associated with the mental state of their family members. When a caregiver’s mental state is “poor”, the patient is more likely to relapse, which could lead a prolongation of recovery. Agreeing with this hypothesis, it is recommended to consider the mental well-being of the informal carers by evaluating patient recovery time [22].

The patient–caregiver dyad has also been studied in the traumatological and chronic fields. It has been well documented in hip injuries, while less attention is paid to other orthopedic conditions. Quantitatively, 19 studies deal with hip fractures, four studies deal with knee injuries, five studies with spine injuries, and two studies with general orthopedic pathologies. All revised studies refer to the caregiver strain and the amount of care for the patient with the orthopedic condition, which will reduce the quality of life of the caregiver due to the resulting growing stress. Many studies focus on patient outcomes related to care by a caregiver. The aim is to improve care but also to reduce the strain and stress factors attributable to this type of relationship established with the patient.

The present results should be interpreted in the context of the strengths and weaknesses of the studies composing the orthopedic caregiver. For example, most studies of orthopedic caregivers have used self-report questionnaires rather than assess the level of quality of life or have used rating scales that emphasize increasing caregiver strain.

Self-reported quality of life was greater than objective measures of quality of life, so it is possible that the actual caregivers’ stress level was even worse than that estimated by the present systematic review. Repeating objective measures for all the caregivers would yield more accurate estimates of the real caregiver strain when caring for orthopedic patients.

Caregivers could be important to patients’ healthcare, particularly according to the duration of caregiving, workload and stress level: these are also the factors that increase caregiver difficulties. Patients demonstrated the greatest increase in quality of recovery thanks to communication with their caregiver and thanks to the help they received from caregivers in maintaining social interaction. Besides, caregivers could improve orthopedic patients’ life with behavioral interventions such as emotional comfort and support, both during pre-operative and post-operative periods.

### Limitations

There were several limitations to this study. Further research is needed to examine how the intervention described would be successful with a larger sample. The study lacked a control group: future research is needed using a design that randomly assigns participants to intervention and control groups. Despite these limitations, the interventions were successful in increasing knowledge of caregivers’ importance in orthopedic disease.

## 5. Conclusions

Despite the challenges in studying the role of caregivers in orthopedic diseases and family caregiver strain and challenges when caring for orthopedic patients, the literature indicates that not only the increase in caregivers’ stress levels but also the decrease in quality of life was less severe in caregivers who received appropriate behavioral interventions including health care advice, such as medication advice or psychological tips. To improve the quality of health care, stressors should be considered for caregivers due to the high strain, especially in the post-discharge period.

Clinicians should consider the importance of caregiver interventions, not only for the orthopedic patient but also for the spouse, child, or friend who will be providing care for that individual. Further studies should focus on the important physical and mental role of the informal caregiver for patients who receive orthopedic surgery and the importance of psychological sphere for the patient–caregiver dyad. Focusing on caregivers’ welfare rather than only on patient well-being could radically improve both caregivers’ performance and patients’ recovery.

## Figures and Tables

**Figure 1 jcm-09-01497-f001:**
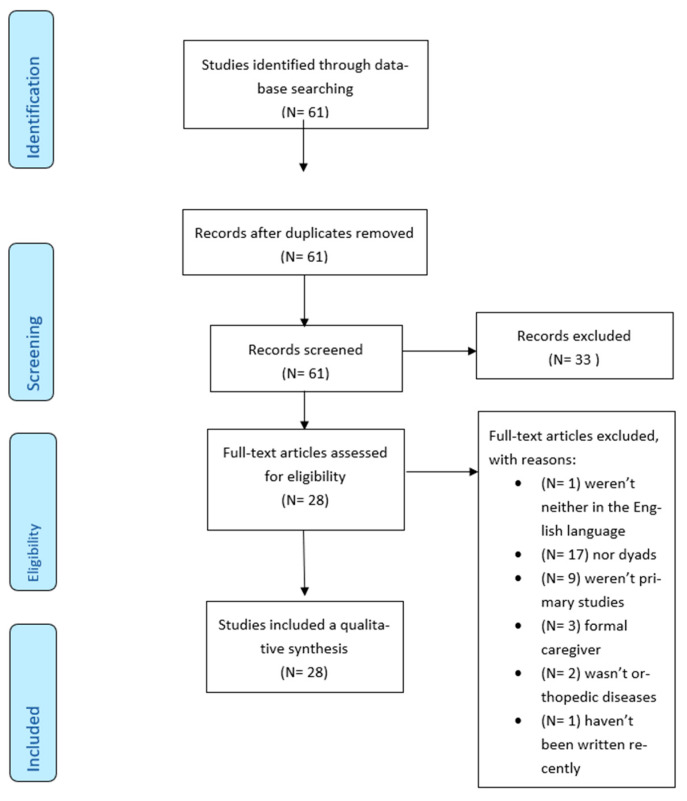
Study selection process and screening according to the PRISMA flow chart [12].

**Table 1 jcm-09-01497-t001:** Characteristics of included studies: author and country; number of patients; number of caregivers; outcome measures; follow-up; data findings; conclusion; relationship with the patients.

Author and Country; Number of Patients; Number of Caregivers	Outcomes Measures; Follow-up	Outcome Measures *; Follow-up	Relationship with the Patients
Shen, 2015, China [13]; 492; 539.	FES-I (*p* < 0.001); FRS (*p* < 0.001); VAS 24 months	75.4% of family caregiver and 70.7% of patients showed FOF; Regular follow-up examinations showed a lower family caregiver FES-I score and a higher patient FRS.	321 spouses (59.5%), 171 (31.7%) offsprings
Patrocinio Ariza-Vega, 2019, Spain [14]; 172; 172.	CSI (*p* < 0.001); 1 year	High level of caregiver difficulties at the hospital, at 1 and 3 months, and at 1 year after surgery; Support and training as strategies of treatment to reduce caregivers’ difficulties.	Partner/spouse 39 (23%), son 15 (8%), daughter 94 (55%), others 24 (14%)
Margaret J. Bull, 2017, US [7]; 39; 39.	CAM (Sensitivity of 94–100%, Specificity of 90–95%); FAM-CAM (Sensitivity of 87.5, Specificity of 94.2%); BSS; 3 weeks pre-surgery; 2-weeks and 2 months post-hospitalization. Caregivers: 2-days post-surgery	The caregiver rating is high on the FAM-CAM 2 days after surgery; Recognizing presence or absence of delirium symptoms by caregivers.	Spouse 24 (62%), daughter 8 (21%), others 7 (17%)
Maria Crotty, 2003, Australia [15]; 32; 66.	MBI (*p* = 0.738); TUG (*p* = 0.001) Medical Outcomes; SF-36 (*p* = 0.689); CSI (*p* = 0.140); 4 months; 12 months	Caregiver difficulty reduction is achieved by home-based therapy and rehabilitation for patients; Functionally independent patients, return home earlier, with increased involvement of caregivers.	NS
Cuicui Li, 2018, China [16]; 87; 87.	ZBI (36.83 ± 13.30); GSE (21.67 ± 7.65); 0	Moderate or severe caregiver difficulties; Social support and self-efficacy might be helpful to reduce caregivers’ difficulties.	Mothers 51 (58.6%), others 36 (41.4%)
Rachel L. Difazio, 2016, US [17]; 44; 44.	CP CHILD (*p* < 0.001); ACEND (*p* = 0.26); 1 year	Children’s HRQOL improved over 12 months after spinal surgery; steady improvement over time after hip surgery, decrease at 6 weeks; Caregivers reported an improvement in HRQOL 1 year after orthopedic surgery.	NS
Mohammad Hossein Ebrahimzadeh, 2013, Iran [18]; 72; 72.	SF-36 (*p* < 0.001); 6 months	Wives worked full time at home. 88.9% of veterans had a paraplegic lesion; The SF-36 scores of the spouses were lower. The caregivers’ challenges can impact the QOL of caregivers.	Spouse 72 (100%)
Jacobi Elliott, 2014, Canada [19]; 8; 11.	Questionnaire; 0	Facilitators and barriers included prior health care experience, trusting relationships, and the rural setting; Effective strategies to improve information sharing and care continuity may be involved.	Adult children 6 (54.5%), spouses 2 (18.1%)
Amit Jain, 2018, US [20]; 251; 251.	HRQOL (*p* < 0.001); 2 years post-surgery RNLI (33.3); CBI	HRQL: 74% of caregivers are a “lot better”; Caregiving in spinal surgery is ranked as the most beneficial intervention in the patients’ lives. Caregiver burden is high at 18 and 24 months post-hip-fracture;	NS
Katherine S. McGilton, 2019, Canada [21]; 76; 76.	2 years post-hip fracture	There is a need for interventions for patients to enhance their RNLI and to support caregivers in decreasing their difficulties in caring.	Marital status, married or common-law partner 26 (34%)
Laura Churchill, 2018, GB [22]; 14; 14.	Questionaire; 6 months Questionnaire;	Concerns and challenges are mobility, pain, self-care and caregiver support; Outpatient THA can be implemented with pre-operative education, clarification of recovery processes and expectations. More details are needed from care providers to self-manage symptoms.	NS
Odom-Forren J, 2017, US [3]; 9; 10.	2 weeks after surgery	Nurses should be focused on preparing patients to manage sustained recovery issues at home.	Spouses 7 (70%), parents 2 (20%)
Jung-Ah Lee, 2014, US [11]; 30; 30.	Questionaire; At hospitalization	Patients and caregivers take daily injections of heparin. Patients with hip fracture and their caregivers may need further VTE preventive education.	child or son/daughter-in-law 19 (63.4%), 3 (10%) spouses
Pi-Chu Lin, 2007, Taiwan [23]; 95; 95.	OMFAQ; SERS; FOS; FFRS; FRS; CBI; 1 week and 1 month after discharge	1 week after hospital discharge the patients’ physical functioning, self-efficacy, and social support contributed to variance in caregivers’ difficulties; A health education and social support program should be designed to improve the primary caregiver’s knowledge and to reduce the burden of care.	Apouses 30 (31.6%), sons 20 (21.1%), 18 (18.9%) daughters, 17 daughters-in-law (17.9%), grandchildren 7 (7.4%)
Hsin-Yun Liu, 2015, Taiwan [24]; 276; 276.	CBI; CMMSE; PS; MICROFET2; MNA; SF-36; 1-3-6-12 months after discharge	MCS levels were lower (22.4%), moderate (34.1%) and highest (43.5%); Health care providers could consider family caregivers’ mental well-being while estimating recovery times and health outcomes of patients.	Spouse 70 (25.3%), son 61 (22.1%), daughter 57 (20.6%), daughter-in-law 71 (25.7%), other 17 (6.1%)
Asha Manohar, 2014, US [25]; 44; 44.	Questionnaire; ADL; Post-operative days; 0-3-7-30	Many patients needed more time to resume their ADL. Primary caregivers’ disturbances were emotional and physical; Informal caregiving may be an unrecognized physical and psychological burden and may have a significant societal impact.	NS
Mariana Ortiz-Piña, 2019, Spain [26]; 70; 70.	FIM; Euro-Qol/EQ-5D; TUG; CBZI; SPPB; ADS; 4 weeks and 12 weeks after discharge	70 patients with a high pre-fracture functional level were allocated into a telerehabilitation group; Telerehabilitation is an option to promote recovery of the pre-fracture functional level.	NS
Mashfiqul A Siddiqui, 2010, Singapore [27]; 76; 76	CSI; 6 months	To 1 week of admission, and at 6 months, the caregivers were stressed. The stress factor was a financial strain. Adequate resources should be available to caregivers of patients with osteoporotic hip fractures.	NS
Benedict U. Nwachukwu, 2019, NS [28]; 95; 95.	UCLA Activity Score; HRQoL; Rehabilitation; 2 years post-surgery	Active adolescents assigned higher utility to achieve a stable return to the same function and lower utility to health states in which they were not fully participating in sport; These findings provide insight into the health-related quality of life impact for acute patella dislocations and their management.	NS
Joshua A. Parry, 2019, NS [29]; 29; 29.	CBI; DS 6 months	Caregivers have negative effects on their finances, relationships, work hours, or intent to place the patient in a care facility. Caregivers with high caregiver burdens were more likely to consider the placement of the patient into a long-term care facility.	NS
Sara Elli, 2018, Italy [30]; 147; 147.	BRASS; At the beginning of the rehabilitation program	The caregivers assign lower scores than the doctor; Caregivers’ altered perceptions can lead to a general lack of satisfaction with the outcome at the end of the rehabilitation process.	NS
Yea-Ing Lotus Shyu, 2012, Taiwan [31]; 135; 151.	PRS; MOS; SF-36; CBI; 1-3-6-12 months after discharge	Caregivers’ mental health was lower at 12 and 1 month after discharge; The home care nurses should develop interventions early after discharge.	1/3 sons (32.66%), daughters-in-law (26.7%), spouses (20%), daughters (14.1%).
Åsa Johansson Stark, 2016, Finland [32]; 306; 306.	During recovery	If nurses gave information to partners, they experienced a greater quality of recovery; Spouses’ emotional state is important in the patients’ quality of recovery.	Spouses 306 (100%).
Justine Toscan, 2012, Canada [33]; 6; 6.	Questionnaire; During transition care	Four factors related to illness were confusion, unclear roles and responsibilities, diluted personal ownership over care, and role strain; Supports the notion of collaborative practice and includes an appropriate, informed role for patients and informal caregivers.	Children 5 (99%)
Cornelis L. P. van de Ree, 2017, Netherland [34]; 123; 123.	CarerQoL; 7D score; 1-3-6 months	The average amount of informal care provided per patient per week was 39.5 during the first six months; The Carer QoL was not associated with the intensity of the provided informal care.	Partners (44.7%), child (43.1%), sibling (5.7%), others (6.5%)
Li-Chu Wu, 2013, Taiwan [35]; 116; 116.	Questionnaire; 1 month after discharge	Impairments in physical functions were standing up/sitting down and dressing. The care needs were wound care, medical visits, cleaning, maintaining living; The physical function status was improved 1 week and after 1 month after discharge. The care needs and the difficulty of tasks for caregivers were negatively related to physical functional status.	Daughters or Sons (54.3%), Spouses (34.5%), Foreign workers (11.0%).
Jayson D. Zadzilka, 2018, US [36]; 150; 150.	CSI; KOOS; 4 weeks and 1 year after surgery	CSI scores at 1 year were lower; The caregivers’ difficulties were high in the early post-operative period. It was close to zero by one year post-operation.	NS

ACEND: Assessment of Caregiver Experience with Neuromuscular Disease; BRASS: Blaylock Risk Assessment Screening Score; BSS: Bakas Satisfaction Scale; Carer QoL 7D Score: Carer Quality of Life 7D Score; CAM: Confusion Assessment Method; CBI: Chinese Barthel Index; CBS: Caregiver Burden Rating Scale; CBZI: Caregiver Burden Zarit Intervention; CMMSE: Chinese Mini Mental Status Examination; CP CHILD: Cerebral Plasty Child; CSI: Caregiver Strain Index; DS: Depression Scale; EQoL-5D: Euro Quality of Life 5D; FAM-CAM: Family Confusion Assessment Method; FES-I: Falls Efficacy Scale-International; FFRS: Family Function Rating Scale; FIM: Functional Indipendence Measure; FOS: Filial Obligation Scale; GSE: General Self-Efficacy Scale: HADS: The Hospital Anxiety and Depression Scale; HRQoL: Health Related Quality of Life; HSS-Pedi-FABS: Hospital for Special Surgery Pediatric Functional Activity Brief Scale; IFS: International Fitness Scale; IADL: Instrumental Activities of Dayly Living; KEso: Knowledge Expectations of Significant Other; KOOS: Knee Injury and Osteoarthritis Outcome Score; MBI: Maslach Burnout Inventory; MCI: Mild Cognitive Impairment; MICRO FET2: Micro Force Evaluation Testing 2; MNA: Mini Nutritional Assesment; MOS: Medical Outcome Study; MOS SF-36: Medical Outcome Study Short Form-36; OMFAQ: Multidimensional Functional Assessment Questionnaire; PRS: Performance Related Scale; PS: Pain Scale; QoL: Quality of Life; RKso: Received Knowledge of Significant Other; RNLI: Reintegration to Normal Living Index; SERS: Self Efficacy Rating Scale; SFES-I: Short Falls Efficacy Scale-International; SF-36: Short Form 36; SPPB: Short Physical Performance Battery; TUG: Timed Up and Go Test; UCLA: University of California, Los Angeles Loneliness Scale; VAS: Visual Analog Scale; VTE: Venous Thromboembolism; ZBI: Zarit Burden Interview.

**Table 2 jcm-09-01497-t002:** Characteristics of orthopedic diseases in included studies.

JOINTS			TOT *	%
**Hip**	Hip fracture Hip arthroplasty Hip deformity	19 1 1	21	75%
**Backbone**	Spinal arthrodesis Scoliosis Spine deformity Cord injury	1 1 1 1	4	14.3%
**Knee**	Knee fracture Knee artrhroplasty Patella fracture	1 1 1	3	10.7%

* TOT: Total.

**Table 3 jcm-09-01497-t003:** Caregivers’ demographic characteristics (relationship with patients).

Relationship with Patients	N	%
**SPOUSE**	726	23.9%
**CHILDREN (DAUGHTERS AND SONS)**	468	15.4%
**OTHERS**	162	5.3%
**DAUGHTER IN LAW**	160	5.2%
**MOTHERS**	51	1.7%
**GRANDCHILDREN**	25	0.8%
**SIBLING**	7	0.2%
**TOTAL**	1599	52.7%
**N/A**	**1435**	**47.2%**

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
