# Peer review of "Family Caregiver Strain and Challenges When Caring for Orthopedic Patients: A Systematic Review"

_jcm, 2020, doi:10.3390/jcm9051497_

Round 1
Reviewer 1 Report
I would like to thank the authors for completing this interesting review of caregiver burden in the context of orthopaedic disease. However, after reading the text I have several queries:
- In the materials and methods section, the search terms used for scouring research databases were listed. While comprehensive, several choices were made that I feel may have limited the number of results that would have been generated. For example the first search term was 'caregiver or spouse' but this doesn't incorporate the term relative (as most caregivers are relatives) which if added may have identified additional studies. Another example is the combination of the terms hip, knee shoulder etc with the term 'caregiver stress' as opposed to the more general inclusive term stress.
- Why were 'caregivers without relation' an excluding factor?
- There is some overlap between tables 1 and 2 and I feel they can be combined. For example the new combined table could have the column headings: Author , Type of study, Outcome measures, follow up, number of carers and patients (displayed as a ratio), findings and Coleman score.
- Figure 1 should be redesigned do the records excluded (n=33) leads on to an explanation of why those records were excluded. They should not be in separate boxes.
- Consideration should be given to reorganising table 3 in terms of closeness of relation to patient. For example Spouse, partner (effectively a spouse), Daughter, son, (can be merged into one possibly) Daughter/son in law (unless no son in laws acted as carers) Mother, Sibling, Grandchildren Other.
- In the results section when talking about follow up , to improve readability the explanation should be organised from the shortest amount of time post op to the longest (i.e. day, week, month year etc)
- In section 3.2 what is meant by 'care burden decrease and functional recover increase in 25% of studies' is that an intervention or a description of what occurred post intervention
- Some discussion points appear in the results section (they should be moved to the discussion) for example part of line 176 and lines 177 and 178
- In the discussion, reference is made to relapse (lines 245-248) . What is meant by relapse? Increase in pain? Decrease in mobility? decrease in mental health? prolongation of recovery?
Author Response
Manuscript ID jcm-775963
Family Caregiver Strain and Challenges When Caring for Orthopedic Patients: A Systematic Review
Many thanks for the opportunity to revise the above manuscript. Please find attached the revised version of the above manuscript. The comments of the Reviewer 1 have been carefully considered and implemented as follows:
Reviewer 1
I would like to thank the authors for completing this interesting review of caregiver burden in the context of orthopaedic disease. However, after reading the text I have several queries:
- In the materials and methods section, the search terms used for scouring research databases were listed. While comprehensive, several choices were made that I feel may have limited the number of results that would have been generated. For example the first search term was 'caregiver or spouse' but this doesn't incorporate the term relative (as most caregivers are relatives) which if added may have identified additional studies. Another example is the combination of the terms hip, knee shoulder etc with the term 'caregiver stress' as opposed to the more general inclusive term stress.
Answer: Many thanks for this comment. We added the term “relative” and the more general inclusive term “stress” in our research as you suggest, but the results generated didn’t change.
- Why were 'caregivers without relation' an excluding factor?
Answer: “Caregivers without relation” were an excluding factor because the aim of the study was to study the role of informal caregivers’ strain and difficulties when caring for orthopaedic patients. We added “informal” in the sentence that explained the aim of the study (line 108)
- There is some overlap between tables 1 and 2 and I feel they can be combined. For example the new combined table could have the column headings: Author, Type of study, Outcome measures, follow up, number of carers and patients (displayed as a ratio), findings and Coleman score.
Answer: Many thanks for your comment. We have modified Table 1:
- We have included a footnote at the bottom of the table which reports the abbreviations in alphabetical order.
- We made three columns (author, number of caregivers and number of patients) as one: author; number of caregivers; number of patients.
- We made outcomes and follow-up columns into one column.
- We made data findings and conclusion columns one column.
- In the outcomes measures column we included quantitative data from studies where available.
- We removed Coleman score column.
- We included percentages in Shen, 2015 relationship with the patients and we placed each type of relationship with commas in between not each on a separate line.
- We condensed words in follow-up column.
We also have modified Table 2 as suggested by Reviewer 2: This table was moved into supplementary material and in part was merged with Table 1. We removed number of caregivers column and number of patients column.
- Figure 1 should be redesigned do the records excluded (n=33) leads on to an explanation of why those records were excluded. They should not be in separate boxes.
Answer: Figure 1 was redesigned adding an explanation of why 33 records were excluded.
- Consideration should be given to reorganising table 3 in terms of closeness of relation to patient. For example Spouse, partner (effectively a spouse), Daughter, son, (can be merged into one possibly) Daughter/son in law (unless no son in laws acted as carers) Mother, Sibling, Grandchildren Other.
Answer: Many thanks for this comment. Table 3 was reorganized in terms of closeness of relation to patient. Spouse and partner are merged into one, as well as Daughter and son which are merged in Children.
- In the results section when talking about follow up, to improve readability the explanation should be organised from the shortest amount of time post op to the longest (i.e. day, week, month year etc).
Answer: We thank the reviewer for his/her suggestion. We improved readability of the follow up section in the results, organizing the explanation from the shortest amount of time post op to the longest, as you suggested.
- In section 3.2 what is meant by 'care burden decrease and functional recover increase in 25% of studies' is that an intervention or a description of what occurred post intervention
Answer: In section 3.2, “care burden decrease and functional recover increase in 25% of the studies” referred to what occurred post interventions.
- Some discussion points appear in the results section (they should be moved to the discussion) for example part of line 176 and lines 177 and 178
Answer: Many thanks for this suggestion. We moved line 176,177 and 178 from result section to the discussion section, as you suggested.
- In the discussion, reference is made to relapse (lines 245-248) . What is meant by relapse? Increase in pain? Decrease in mobility? decrease in mental health? prolongation of recovery?
Answer: In the discussion, the term “relapse” on line 246 means that there could be a prolongation of patient’s recovery. We clarified this in the text.
We thank the Editorial Board for having allowed us to revise our manuscript. I hope that the additions have now improved the paper and that it has now reached the standard necessary to be formally accepted for publication in the Journal of Clinical Medicine.
Yours sincerely,
Umile Giuseppe Longo
Reviewer 2 Report
This paper addresses a highly under-researched topic but one of critical importance. Addressing family care networks has been highlighted as an area of critical research need by the United Nations. However, the way we approach this topic needs to be carefully considered due to the vulnerability of the groups and their stigmatisation. The following areas should be addressed prior to publication.
Major flaws to address:
- Carer stakeholder groups have advocated for the abandonment of the term “burden” as it has been received negatively by family carers. Preferred terms include “strain”, “challenges”, or “difficulties”. We recommend the title and throughout are changed to one of these terms. However, we acknowledge burden must be used when referring to the “zarit burden scale” due to the name of the instrument.
- The method section needs work. Currently there is no clear definition of how the analysis was conducted and there are no definitions of the primary/secondary outcomes. Was the study registered?
- The results section does not adequately address the aim of the study. The outcomes are not clearly analysed (how many studies reported on each outcome, what were the findings) and the data is no reported. Table 1 reports the outcome measures for each study although it does not include the results of these measures to conduct a quantitative analysis.
Other areas to address:
- Punctuation throughout should be consistent, especially in tables.
- Abstract: include in the method section how the analyse was conducted and the MCMS for quality assessment.
- Use past tense throughout the text e.g. ‘are’ to ‘were’.
- Line 20 – reword to “28 articles with 3034 dyads”…
- Line 35 – reword ‘problems come back to their pre-surgery lives’ to ‘problems retuning to surgical lives’.
- Line 40 – ‘ADLs’ not ‘ADLS’
- Line 60 – put reference at end of the sentence.
- Please publish the full search strategy as supplementary material.
- Line 71-75 –
- Do not number the inclusion criteria, report in paragraph form.
- Do not need to include ‘types of’, just report participants, intervention and outcomes.
- Report which outcomes are the primary and secondary outcomes, it is unclear.
- Line 83-84 – reword to ‘search strategies were checked… The exclusion criteria included: … patients and caregivers without relation’.
- Line 86 – reword to ‘Two researchers (VA and VA)…’ and report the reviewers consistently in this format throughout the text.
- Line 87 – ‘met the inclusion…’ not ‘meet’. Reword ‘disagreements have been resolved’ to ‘disagreements were resolved…’
- Line 93 – reword sentence.
- Line 94 – remove ‘in case of’ to ‘Disagreements regarding the exclusion and inclusion criteria were decided by the senior reviewer (VD)’
- Line 97 – reword to ‘The studies were critiqued…’
- Line 98 – include reference for MCMS. Reword ‘evaluate its quality’ to ‘evaluate the quality’
- Line 100 – reword ‘the study is reliable and deserves to be considered as valid’ to ‘the study is reliable and is considered valid’
- Quality assessment: consider if you can conduct a GRADE assessment for the quality of evidence for the primary outcome/s. If this is unable, report why.
- Definition of outcomes: need to report what the outcomes were including the primary and secondary outcomes before reporting results in tables. Line 104 –reword ‘have been’ to ‘were’, report which review in brackets and end sentence after ‘…study analysis’ Remove line 105-106.
- Move all of the tables and figures from the method section into the results section. Each table and figure title need to provide more detail.
- Include a table in supplementary material of the quality assessment results of how the reviewers reached the Coleman score for each study. Remove the Coleman score column from Table 1 and insert in the new table.
- Table 1: This table is too big for the body of the text and can easily be condensed. Consider the following:
- Include a footnote at the bottom of the table which reports the abbreviations in alphabetical order.
- Make three columns (author, number of caregivers and number of patients) as one e.g. author; number of caregivers; number of patients.
- Make outcomes and follow-up columns into one column.
- Make data findings and conclusion columns one column.
- In the data findings column add subheadings for the outcomes (quality of life, increasing stress factors, etc.) to make it clearer to read and include quantitative data from studies where available e.g. quality of life average VAS score.
- Remove Coleman score column as stated above.
- Include percentages in Shen, 2015 relationship with the patients and place each type of relationship with commas in between not each on a separate line.
- Condense words in follow-up column e.g. 3-weeks pre-surgery, 2-weeks and 2-months post-surgery.
- Line 108 – include which reviewer in brackets.
- Line 109 – reword ‘who have been’ to ‘which were’ and reword next sentence to ‘They rated the patients perception of…’
- Line 115-116 – put in methods quality assessment paragraph.
- Line 118-119 – restructure sentences, first sentence should be in results section. Second sentence is confusing. Consider ‘Data was extracted into Microsoft and Word and Excel tables (reference).
- Table 2: This table can be moved into supplementary material or merged with Table 1. There is repeated data reported in this table which should be removed (number of caregivers, number of patients). Report information consistently – if the aim of each study starts with a capital letter, report all this way.
- Table 3: Consider not centring the words in the first column. Report percentages with decimals not commas. Reword ‘totale’ to ‘total’.
- Figure 1: 61 studies from four databases is a very low number of studies retrieved. There is a possibility there was a flaw in the search and should be reported in the limitations.
- Are the records excluded (n=33) and full-text articles exclusion reasons reporting the same thing? Both equal 33 studies, combine if so. The numbers do not add up in this figure - full-text articles eligible (n=28) and studies included (n=28) is incorrect. Correct these errors.
- Reword the reasons for exclusion e.g. ‘weren’t neither in the English language’ to ‘non-English studies’ or combine with ‘weren’t primary studies’ and ‘haven’t been written recently’ as ‘study design’. ‘Nor dyads’ to ‘population’. ‘Wasn’t orthopaedic diseases’ to ‘intervention’.
- Line 124 – reword to ‘The main data extracted and summarised in Table 2 included: age, authors, study design, number of caregivers and patients recruited, orthopaedic disease type and the aim of the study’. This paragraph describing the tables should be moved into the results section.
- Results: report the search results sentences in a paragraph with the study and patient characteristics. Insert Figure 1 after this paragraph.
- Line 131 – reword to ‘After de-duplication, title/abstract and full-text review, 28 studies were eligible for the review and quality assessment’.
- Line 137-152 need to be reworded, currently many sentences are confusing to the reader. E.g. line 151-152 reword to ‘Fourteen percent of studies reported caregiver burden lasted for one-year, 10.7% for six-months, 7.1% for one-month and 3.5% for two-years’.
- Line 142-143 – report number of months, weeks, days not ‘few’.
- Table 4: report the percentages to one decimal and do not use commas. This should be consistent throughout the review. Include a footnote at the bottom of the table for abbreviations (TOT).
- Line 162 – reword ‘to measuring empowering’ to ‘to measure empowering’
- Line 164 – reword to ‘Measures for general health status included: the Short Form 36 Health Survey and physical performance was evaluated using…’
- Line 168 – italics for ‘general’
- Line 170 – reword to ‘The ADLs for patients and caregivers were used using….’
- Line 172 – reword sentence.
- Line 174 – reword to ‘Other rating scales are reported in Table 2 without…’
- Line 176 – comma after analysed.
- Line 180 – remove first sentence. Reword second sentence to ‘The average MCMS for the 28 included studies was 64.9.’
- Line 182-185 – these sentences are confusing and unsure if this information is needed.
- The role of caregivers in orthopaedics diseases section should be under the study characteristics section in results as this is the main aim of the study.
- Line 187-190 – reword to ‘The most common relationships between the primary caregiver and the care recipient in this review included spouse (22.1%), daughter (8.7%), son (6.7%), daughter-in-law (5.2%) and others including partner, mothers, grandchildren and sibling (9.2%) (Table 3).
- Line 191 – reword to ‘Two studies identified a significant…’ and describe what the relationship was.
- Line 194-197 – These sentences are confusing, reword and include references of which studies the sentences are referring to.
- Line 215 – reword sentence.
- Discussion: ensure the results reported in the discussion were reported in the results section first. The discussion does not include limitations of the review but limitations to some included studies.
- Line 234 – reword sentence to ‘This review showed the importance of information and education before….’
Author Response
Manuscript ID jcm-775963
Family Caregiver Strain and Challenges When Caring for Orthopedic Patients: A Systematic Review
Many thanks for the opportunity to revise the above manuscript. Please find attached the revised version of the above manuscript. The comments of the Reviewer 2 have been carefully considered and implemented as follows:
I would like to thank the authors for completing this interesting review of caregiver strain in the context of orthopaedic disease. However, after reading the text I have several queries:
For the major flaws to address
- Carer stakeholder groups have advocated for the abandonment of the term “burden” as it has been received negatively by family carers. Preferred terms include “strain”, “challenges”, or “difficulties”. We recommend the title and throughout are changed to one of these terms. However, we acknowledge burden must be used when referring to the “zarit burden scale” due to the name of the instrument.
Answer: Many thanks for this comment. We replaced the term “burden” all over the manuscript with the terms suggested: “strain”, “challenges”, or “difficulties”.
- The method section needs work. Currently there is no clear definition of how the analysis was conducted and there are no definitions of the primary/secondary outcomes. Was the study registered?
Answer: Thank you for pointing this. We revised the method section adding primary and secondary outcomes.
The request of study registration has been sent to PROSPERO and we are waiting response.
- The results section does not adequately address the aim of the study. The outcomes are not clearly analyzed (how many studies reported on each outcome, what were the findings) and the data is no reported. Table 1 reports the outcome measures for each study although it does not include the results of these measures to conduct a quantitative analysis.
Answer: Many thanks for your comment. We have modified the results section. Outcomes measures, quantitative results and conclusion are shown in Table 1. The aim of each study could be found in Table 2.
- Punctuation throughout should be consistent, especially in tables
Answer: punctuation was corrected throughout the text
- Abstract: include in the method section how the analyse was conducted and the MCMS for the quality assessment.
Answer: We thank the reviewer for his/her suggestion. We added in the method section of the abstract that the analysis was conducted according to the PRISMA Guidelines and the MCMS for the quality assessment.
- Use past tense throughout the text e.g. ‘are’ to ‘were’.
Answer: We thank the reviewer for his/her suggestion. The past tens were used throughout the text.
- Line 20 – reword to “28 articles with 3034 dyads”
Answer: We thank the reviewer for his/her suggestion. The sentence “28 articles with 3034 dyads” was reworded in “28 studies were included in the systematic review, about these studies were analyzed 3034 dyads”.
- Line 35 – reword ‘problems come back to their pre-surgery lives’ to ‘problems returning to surgical lives’.
Answer: We thank the reviewer for his/her suggestion. The sentence was reworded as required.
- Line 40 – ‘ADLs’ not ‘ADLS’
Answer: We thank the reviewer for his/her suggestion. The word was reworded as required.
- Line 60 – put the reference at the end of the sentence.
Answer: We thank the reviewer for his/her suggestion. The reference was placed at the end of the sentence according to indication.
- Please publish the full search strategy as supplementary material.
Answer: We thank the reviewer for his/her suggestion. The full search strategy was published as supplementary material according to indication.
- Line 71-75 –Do not number the inclusion criteria, report in paragraph form.
Answer: We thank the reviewer for his/her suggestion. The inclusion criteria were reported in paragraph form.
- Do not need to include ‘types of’, just report participants, intervention and outcomes.
Answer: We thank the reviewer for his/her suggestion. It was modified as required.
- Report which outcomes are the primary and secondary outcomes, it is unclear.
Answer: We thank the reviewer for his/her suggestion. Primary and secondary outcomes were clarified.
- Line 83-84 - reword to ‘search strategies were checked… The exclusion criteria included: … patients and caregivers without relation’.
Answer: the sentence was reworded according to indication
- Line 86 – reword to ‘Two researchers (VA and VA)…’ and report the reviewers consistently in this format throughout the text.
Answer: We thank the reviewer for his/her suggestion. It was modified as required.
- Line 87 – ‘met the inclusion…’ not ‘meet’. Reword ‘disagreements have been resolved’ to ‘disagreements were resolved…’
Answer: We thank the reviewer for his/her suggestion. The sentences were reworded as indicated.
- Line 93 – reword sentence
Answer: We thank the reviewer for his/her suggestion.Tthe sentence “The study included, according to inclusion criteria, have been achieved from 2003 to 2019” was reworded in “the studies included articles published from 2003 to 2019”.
- Line 94 – remove ‘in case of’ to ‘Disagreements regarding to the exclusion and inclusion criteria were decided by the senior reviewer (VD)’.
Answer: We thank the reviewer for his/her suggestion. The sentences were corrected as required.
- Line 97 – reword to ‘The studies were critiqued…’
Answer: We thank the reviewer for his/her suggestion. The sentence was corrected as indicated.
- Line 98 – include reference for MCMS. Reword ‘evaluate its quality’ to ‘evaluate the quality’
Answer: We thank the reviewer for his/her suggestion. The reference for MCMS was included. The sentence was corrected as indicated.
- Line 100 – reword ‘the study is reliable and deserves to be considered as valid’ to ‘the study is reliable and considered valid’
Answer: We thank the reviewer for his/her suggestion. The sentence was corrected as indicated.
- Quality assessment: consider if you can conduct a GRADE assessment for the quality of evidence for the primary outcome/s. it this is unable, report why.
Answer: thank you for the suggestion, but we are unable to conduct a GRADE because the GRADE assess the strength of recommendations using two categories (for or against an option) and the studies analyzed in our review didn’t include two intervention groups (https://gradeworkinggroup.org). However, outcome was correlated with the total Coleman Methodology Score to assess the impact of methodology on the reported outcomes. Finally, the Coleman Methodology Score was correlated with the level-of-evidence rating.
- Definition of outcomes: need to report what the outcomes were including the primary and secondary outcomes before reporting results in tables. Line 104 – reword ‘have been’ to ‘were’, report which review in brackets and end sentence after ‘…study analysis’. Remove line 105-106.
Answer: the primary and secondary outcomes were included as indicated. The tense was changed in “were” as indicated. The reviewers were reported in brackets at the end of the sentence after “…study analysis”. Line 105-106 were removed as indicated.
- Move all of the tables and figures from the method section into the results section. Each table and figure title need to provide more detail.
Answer: the tables and figures were moved into the results section. Table and figure titles were reported in detail.
- Include a table in supplementary material of the quality assessment results of how the reviewers reached the Coleman score for each study. Remove the Coleman score column from Table 1 and insert in the new table.
Answer: the table for Coleman score quality assessment results was added in supplementary material. The Coleman score column was removed from Table 1 as indicated.
- Table 1: this table is too big for the body of the text and can easily be considered. Consider the following:
- Include a footnote at the bottom of the table which reports the abbreviations in alphabetical order.
Answer: a footnote at the bottom of the table 1 which reports the abbreviations in alphabetical order was included as indicated.
- Make three columns (author, number of caregivers and number of patients) as one e.g. author; number of caregivers; number of patients.
Answer: the first columns in table 1 reports author, number of caregivers and number of patients as indicated
- Make outcomes and follow-up columns into one column.
Answer: outcome measures and follow-up were unified in one column in table 1 as indicated.
- Make data findings and conclusion columns one column.
Answer: data findings and conclusion were unified in one column in table 1 as indicated.
- In the data findings column add subheadings for the outcomes (quality of life, increasing stress factors, etc.) to make it clearer to read and include quantitative data from studies where available e.g. quality of life average VAS score.
Answer: thanks for your suggestion, in the data finding column several details were reported, that clarify the read. The quantitative data concern the score of the rating scales. They were included in the outcome measures and follow-up period column.
- Remove Coleman score column as stated above.
Answer: Coleman score column was removed and it was reported in supplementary material as indicated.
- Include percentages in Shen, 2015 relationship with the patients and place each type of
relationship with commas in between not each on a separate line.
Answer: the percentages in Shen, 2015 relationship was included. Each type of relationship was reported in the column with commas and not on a separate line.
- Condense words in follow-up column e.g. 3-weeks pre-surgery, 2-weeks and 2-months post-surgery.
Answer: the words indicated were condensed in follow-up and outcome measures column.
- Line 108 – include which reviewer in brackets
Answer: (VA and VA) reviewer were included in brackets as indicated.
- Line 109 – reword ‘who have been’ to ‘which were’ and reword next sentence to ‘They rated the patients perception of…’
Answer: the sentence was reworded as indicated.
- Line 115-116 – put in methods quality assessment paragraph.
Answer: quality assessment paragraph was moved on methods section.
- Line 118-119 – restructure sentences, first sentence should be in results section. Second sentence is confusing. Consider ‘Data was extracted into Microsoft and Word and Excel tables (reference).
Answer: the sentence “Data synthesis have been illustrated in three different Table (1-2-3) and PRISMA flow chart (Figure 1)” was moved in results section. The second sentence was modified as indicated. The reference was inserted.
- Table 2: This table can be moved into supplementary material or merged with Table 1. There is repeated data reported in this table which should be removed (number of caregivers, number of patients). Report information consistently – if the aim of each study starts with a capital letter, report all this way.
Answer: We thank the reviewer for his/her suggestion. Table 2 was added in supplementary material. The capital letter was placed at the beginning of the sentence of each study in “aim of study” column.
- Table 3: Consider not centring the words in the first column. Report percentages with decimals not commas. Reword ‘totale’ to ‘total’.
Answer: corrections were reported in table 3 as indicated
- Figure 1: 61 studies from four databases is a very low number of studies retrieved. There is a possibility there was a flaw in the search and should be reported in the limitations.
Answer: thank you for your opinion. It was reported in limitations section
- Are the records excluded (n=33) and full-text articles exclusion reasons reporting the same thing? Both equal 33 studies, combine if so. The numbers do not add up in this figure - full-text articles eligible (n=28) and studies included (n=28) is incorrect. Correct these errors.
Answer: the records excluded (n=33) and full-text articles exclusion reasons were combined in the same box. Full-text articles eligible (n=28) and studies included (n=28) were corrected and combined in the same box.
- Reword the reasons for exclusion e.g. ‘weren’t neither in the English language’ to ‘non-English studies’ or combine with ‘weren’t primary studies’ and ‘haven’t been written recently’ as ‘study design’. ‘Nor dyads’ to ‘population’. ‘Wasn’t orthopaedic diseases’ to ‘intervention’
Answer: exclusion reasons were reviewed as indicated
- Line 124 - reword to ‘The main data extracted and summarized in Table 2 included: age, authors, study design, number of caregivers and patients recruited, orthopaedic disease type and the aim of the study’. This paragraph describing the tables should be moved into the results section.
Answer: the sentence was reworded and the paragraph describing the tables was moved into the result section as indicated
- Results: report the search results sentences in a paragraph with the study and patient characteristics. Insert Figure 1 after this paragraph.
Answer: the search results sentences were moved on the study and patient characteristics paragraph.
Figure 1 was inserted after this paragraph as indicated.
- Line 131 – reword to ‘After de-duplication, title/abstract and full-text review, 28 studies were eligible for the review and quality assessment’.
Answer: the sentence was reworded as indicated.
- Line 137-152 need to be reworded, currently many sentences are confusing to the reader. E.g. line 151-152 reword to ‘Fourteen percent of studies reported caregiver burden lasted for one-year, 10.7% for six-months, 7.1% for one-month and 3.5% for two-years’.
Answer: the paragraph indicated was reworded clearly. You can find the correction in the paragraph “study and patient characteristic” in the results section. Line 151-152 were reworded as indicated.
- Line 142-143 – report number of months, weeks, days not ‘few’.
Answer: number of months, weeks and days were reported.
- Table 4: report the percentages to one decimal and do not use commas. This should be consistent throughout the review. Include a footnote at the bottom of the table for abbreviations (TOT).
Answers: commas on percentages reported in table 4 and throughout the review were removed. TOT was replaced with “Total”.
- Line 162 – reword ‘to measuring empowering’ to ‘to measure empowering’
Answer: the sentence was replaced as indicated.
- Line 164 – reword to ‘Measures for general health status included: the Short Form 36 Health Survey and physical performance was evaluated using…’
Answer: the sentence was replaced as indicated.
- Line 168 – italics for ‘general’
Answer: the word was replaced as indicated.
- Line 170 – reword to ‘The ADLs for patients and caregivers were used using….’
Answer: the sentence was replaced as indicated.
- Line 172 – reword sentence.
Answer: the sentence was reformulated as “Visual Analogue Scale for Pain (VAS) was the scale most used to evaluate symptoms and Depression scales”.
- Line 174 – reword to ‘Other rating scales are reported in Table 2 without…’
Answer: the sentence was reworded as indicated
- Line 176 – comma after analyzed.
Answer: comma after “analyzed” was introduced as indicated.
- Line 180 – remove first sentence. Reword second sentence to ‘The average MCMS for the 28 included studies was 64.9.’
Answer: We thank the reviewer for his/her suggestion. The first sentence was removed as indicated. The second sentence was reworded and moved on quality assessment section.
- Line 182-185 – these sentences are confusing and unsure if this information is needed.
Answer: We thank the reviewer for his/her suggestion. We reworded the sentence.
- The role of caregivers in orthopaedics diseases section should be under the study characteristics section in results as this is the main aim of the study.
Answer: We thank the reviewer for his/her suggestion. The role of caregivers in orthopaedics disease was moved and combined with the study and patient characteristic section.
- Line 187-190 – reword to ‘The most common relationships between the primary caregiver and the care recipient in this review included spouse (22.1%), daughter (8.7%), son (6.7%), daughter-in-law (5.2%) and others including partner, mothers, grandchildren and sibling (9.2%) (Table 3).
Answer: this sentence was reworded as indicated
- Line 191 – reword to ‘Two studies identified a significant…’ and describe what the relationship was.
Answer: the sentence was reworded and the relationship was specified between caregiver strain, age and gender.
- Line 194-197 – These sentences are confusing, reword and include references of which studies the sentences are referring to.
Answer: these sentences was reworded and moved on study and patient characteristic sections. References of the study were added as indicated.
- Line 215 – reword sentence.
Answer: sentence was reworded in “Physical outcomes on patients and their functional recovery, and also the impact of caregiver assistance on care transition have been studied more frequently”
- Discussion: ensure the results reported in the discussion were reported in the results section first. The discussion does not include limitations of the review but limitations to some included studies.
Answer: the results reported in the discussion were reported in results section as indicated. Limitation paragraph of this systematic review was added in discussion section.
- Line 234 – reword sentence to ‘This review showed the importance of information and education before….’
Answer: We thank the reviewer for his/her suggestion. The sentence was reworded as indicated.
We thank the Editorial Board for having allowed us to revise our manuscript. I hope that the additions have now improved the paper and that it has now reached the standard necessary to be formally accepted for publication in the Journal of Clinical Medicine.
Yours sincerely,
Umile Giuseppe Longo